# Anticipation of temporally structured events in the brain

**Caroline S Lee[1,2], Mariam Aly[1], Christopher Baldassano[1]***

[1]Columbia University, Department of Psychology, New York, United States; [2]Dartmouth College, Department of Psychological and Brain Sciences, Hanover, United States

**Abstract** Learning about temporal structure is adaptive because it enables the generation of expectations. We examined how the brain uses experience in structured environments to anticipate upcoming events. During fMRI (functional magnetic resonance imaging), individuals watched a 90 s movie clip six times. Using a hidden Markov model applied to searchlights across the whole brain, we identified temporal shifts between activity patterns evoked by the first vs. repeated viewings of the movie clip. In many regions throughout the cortex, neural activity patterns for repeated viewings shifted to precede those of initial viewing by up to 15 s. This anticipation varied hierarchically in a posterior (less anticipation) to anterior (more anticipation) fashion. We also identified specific regions in which the timing of the brain's event boundaries was related to those of human-labeled event boundaries, with the timing of this relationship shifting on repeated viewings. With repeated viewing, the brain's event boundaries came to precede human-annotated boundaries by 1–4 s on average. Together, these results demonstrate a hierarchy of anticipatory signals in the human brain and link them to subjective experiences of events.

## Introduction

A primary function of the brain is to adaptively use past experience to generate expectations about events that are likely to occur in the future (*Clark, 2013*; *Friston, 2005*). Indeed, anticipation and prediction are ubiquitous in the brain, spanning systems that support sensation, action, memory, motivation, and language (*den Ouden et al., 2010*). For example, the visual system takes advantage of the world's relative stability over space and time to anticipate upcoming input (*de Lange et al., 2018*; *Summerfield and Egner, 2009*). The majority of studies examining anticipatory signals, however, have tested anticipation based on memory for relatively simple associations between pairs of discrete stimuli, such as auditory tones, lines, dots, oriented gratings, or abstract objects (e.g., *Alink et al., 2010*; *Gavornik and Bear, 2014*; *Hindy et al., 2016*; *Kok et al., 2012*; *Kok et al., 2014*; *Kok and Turk-Browne, 2018*). These studies have found anticipatory signals about a single upcoming stimulus in a variety of brain regions, from perceptual regions (*Kok et al., 2012*) to the medial temporal lobe (*Hindy et al., 2016*; *Kok and Turk-Browne, 2018*). How does the brain use repeated experience in naturalistic environments to anticipate upcoming sequences of events that extend farther into the future?

Prior work has shown that the brain integrates information about the recent past over a hierarchy of timescales (*Aly et al., 2018*; *Hasson et al., 2015*; *Kurby and Zacks, 2008*). Lower-order areas primarily represent the current moment, whereas higher-order areas are sensitive to information from many seconds or even minutes into the past. Higher-order regions with longer timescales play a critical role in organizing perceptual input into semantically meaningful schematic representations (*Baldassano et al., 2017*; *Baldassano et al., 2018*). What is less clear is whether this hierarchy also exists in a prospective direction: as we move from lower-order perceptual systems into higher-order areas, do these regions exhibit different timescales of anticipation into the future? We previously

*For correspondence:
c.baldassano@columbia.edu

Competing interests: The authors declare that no competing interests exist.

**eLife digest** Anticipating future events is essential. It allows individuals to plan and prepare what they will do seconds, minutes, or hours in the future. But how the brain can predict future events in both the short-term and long-term is not yet clear. Researchers know that the brain processes images or other sensory information in stages. For example, visual features are processed from lines to shapes to objects, and eventually scenes. This staged approach allows the brain to create representations of many parts of the world simultaneously.

A similar hierarchy may be at play in anticipation. Different parts of the brain may track what is happening now, and what could happen in the next few seconds and minutes. This would provide a way for the brain to forecast upcoming events in the immediate, near, and more distant future at the same time.

Now, Lee et al. show that the regions in the back of the brain anticipate the immediate future, while longer-term predictions are made in brain regions near the front. In the experiments, study participants watched a 90-second clip of the movie 'The Grand Budapest Hotel' six times while undergoing functional magnetic resonance imaging (fMRI). Then, Lee et al. used computer modeling to compare the brain activity captured by fMRI during successive viewings. This allowed the researchers to watch participants' brain activity moment-by-moment.

As the participants repeatedly watched the movie clip, their brains began to anticipate what was coming next. Regions near the back of the brain like the visual cortex anticipated events in the next 1 to 4 seconds. Areas in the middle of the brain anticipated 5 to 8 seconds in the future. The front of brain anticipated 8 to 15 seconds into the future. Lee et al. show that many parts of the brain work together to predict the near and more distant future. More research is needed to understand how this information translates into actions. Learning more may help scientists understand how diseases or injuries affect people's ability to plan and respond to future events.

found that higher-order regions did exhibit anticipatory signals when individuals had prior knowledge of the general structure of a narrative (*Baldassano et al., 2017*). But these individuals only had knowledge of information at relatively long timescales (e.g., the general sequence of events, and not moment-by-moment perceptual features), so we were unable to assess whether they could generate expectations across the timescale hierarchy.

Here, we examine how the brain anticipates event boundaries in familiar sequences of actions. We used a naturalistic narrative stimulus (a movie), in which regularities are present at multiple timescales. For example, upon second viewing of a movie, one can anticipate the next action to be taken in a given scene, the next character to appear, the next location that is visited, and the last scene of the movie. The presence of predictability at multiple timescales in the same stimulus enables us to identify varying timescales of anticipation in the brain that co-exist simultaneously. We hypothesized that the timescale of anticipation in the brain would vary continuously, with progressively higher-order regions (e.g., prefrontal cortex) anticipating events that are further in the future compared to lower-order regions (e.g., visual cortex).

To test this, we examined brain activity with functional magnetic resonance imaging (fMRI) while individuals watched a 90 s clip from the movie *The Grand Budapest Hotel* six times. To uncover anticipation in the brain, we used a searchlight approach in which, for each region of interest, we fit a hidden Markov model (HMM) to identify temporal shifts between multivariate activity patterns (functionally hyperaligned across individuals using the shared response model [SRM]) evoked by the first viewing of the movie clip compared to repeated viewings (*Figure 1*). This model assumes that the brain's response to a structured narrative stimulus consists of a sequence of distinct, stable activity patterns that correspond to event structure in the narrative (*Baldassano et al., 2017*). We could then identify, on a timepoint-by-timepoint basis, the extent to which viewers were activating event-specific brain activity patterns earlier in subsequent viewings of the movie, by drawing on their prior experience. Because the HMM infers a probability distribution over states, it is able to detect subtle shifts between viewings; activity patterns may reflect a combination of current and upcoming events, and the degree of anticipation can vary throughout the clip.

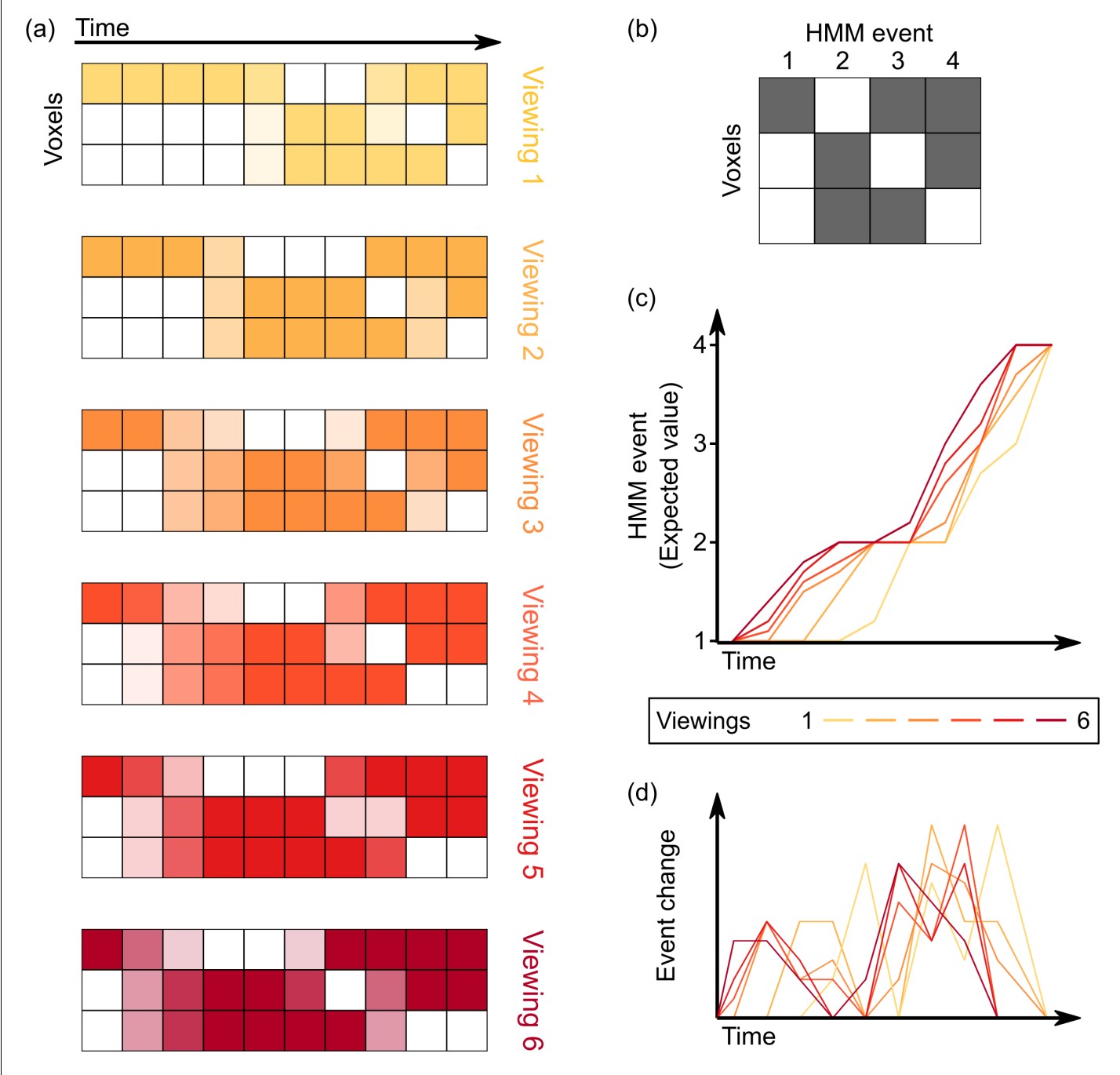

**Figure 1.** Computing varying timescales of anticipatory signals by examining temporal shifts in events across multiple viewings of a movie. (a) Given the voxel by time pattern of responses evoked by the movie clip on each viewing (darker colors indicate higher levels of activity), our goal is to model all viewings as a series of transitions through a shared sequence of event patterns. (b–c) By fitting a hidden Markov model (HMM) jointly to all viewings, we can identify this shared sequence of event patterns, as well as a probabilistic estimate of event transitions. Regions with anticipatory representations are those in which event transitions occur earlier in time for repeated viewings of a stimulus compared to the initial viewing, indicated by an upward shift on the plot of the expected value of the event at each timepoint. (d) Taking the temporal derivative of the event timecourse plot in (c) produces a measure of the strength of event shifts at each moment in time, allowing for comparison with event boundary annotations from human observers.

We also compared the brain's event boundaries (identified by the HMM) to subjective event boundary annotations made by a separate group of participants. This allowed us to test how the relationship between the brain's events and subjective event boundaries changes with repeated viewings. Together, this approach allowed us to characterize the nature of hierarchical anticipatory signals in the brain and link them to behavioral measures of event perception.

## Results

### Timescales of anticipation in the brain

To identify anticipatory signals in the brain, we examined TR-by-TR brain activity patterns during each of the six viewings of the movie clip. For each spherical searchlight within the brain volume, we fit an HMM jointly to all repetitions, to identify a sequence of event patterns common to all viewings and the timing of spatial pattern changes for each viewing. At each timepoint for each viewing, the HMM produced a probability distribution that describes the mixture of event patterns active at that timepoint. Computing the expected value of this distribution provides an index of how the brain transitions through event patterns on each viewing, allowing us to identify how this timing shifts within each region of the brain.

Our analysis revealed temporal shifts in event patterns in many brain regions, including lateral occipital cortex, angular and supramarginal gyri, lateral and anterior temporal lobe, lateral and medial prefrontal cortex (mPFC), and insular cortex (*Figure 2*). The magnitude of this shift varied along a posterior-to-anterior temporal hierarchy (Spearman's rho = 0.58, p=0.0030), with the most anterior regions in the temporal pole and prefrontal cortex showing shifts of up to 15 s on subsequent viewings compared to the first viewing. This hierarchy persisted even when computed on the unthresholded anticipation map including voxels that did not meet the threshold for statistical significance (Spearman's rho = 0.42, p=0.0028; see *Figure 2—figure supplement 1*). There were no significant correlations with the left-to-right axis (rho = 0.06, p=0.41 for thresholded map; rho = 0.12, p=0.29 for unthresholded map) or the inferior-to-superior axis (rho = 0.07, p=0.28 for thresholded map; rho = −0.11, p=0.73 for unthresholded map). We obtained a similar map when comparing the first viewing to just the sixth viewing alone (see *Figure 2—figure supplement 2*).

We compared how this hierarchy of anticipation timescales related to the intrinsic processing timescales in each region during the initial viewing of the movie clip. Identifying the optimal number of HMM events for each searchlight, we observed a timescale hierarchy similar to that described in previous work, with faster timescales in sensory regions and slower timescales in more anterior regions (*Figure 2—figure supplement 3a*). Regions with longer intrinsic timescales also showed a greater degree of anticipation with repeated viewing (*Figure 2—figure supplement 3b*).

We also compared these results to those obtained by using a simple cross-correlation approach, testing for a fixed temporal offset between the responses to initial and repeated viewing. This approach did detect significant anticipation in some anterior regions, but was much less sensitive than the more flexible HMM fits, especially in posterior regions (*Figure 2—figure supplement 4*).

### Relationship with human-annotated events

Our data-driven method for identifying event structure in fMRI data does not make use of information about the content of the stimulus, leaving open the question of how the HMM-identified event boundaries correspond to subjective event transitions in the movie. One possibility is that the brain's event boundaries could *start* well-aligned with event boundaries in the movie and then shift earlier (indicating anticipation of upcoming stimulus content). Alternatively, they may initially lag behind stimulus boundaries (reflecting a delayed response time on initial viewing) and then shift to become *better* aligned with movie scene transitions on repeated viewings. Finally, both patterns may exist simultaneously in the brain, but in different brain regions.

We asked human raters to identify event transitions in the stimulus, labeling each 'meaningful segment' of activity (*Figure 3*). To generate a hypothesis about the strength and timing of event shifts in the fMRI data, we convolved the distribution of boundary annotations with a hemodynamic response function (HRF) as shown in *Figure 4*. We then explored alignment between these human-annotated event boundaries and the event boundaries extracted from the brain response to each viewing, as shown in *Figure 1d*. In each searchlight, we cross-correlated the brain-derived boundary

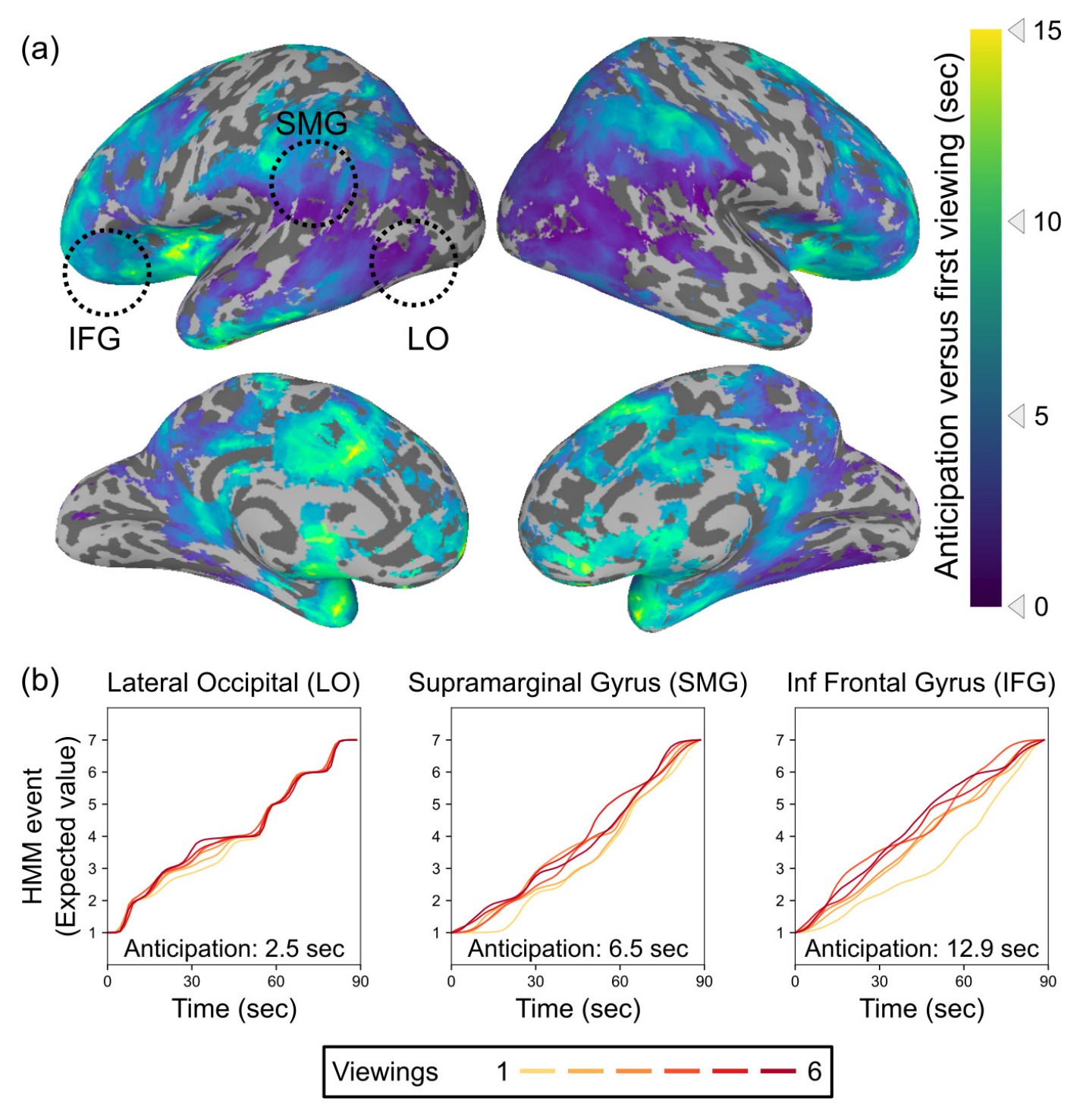

**Figure 2.** Timescales of anticipation vary across the cortical hierarchy. (a) Multiple regions exhibited shifts in event timing between initial and repeated viewings, with event transitions shifting earlier in time with subsequent viewings. Across the brain, anticipation timescales varied from a few seconds to 15 s, with the longest timescale anticipatory signals in prefrontal cortex and the temporal pole. Anticipation followed a posterior-to-anterior hierarchy, with progressively anterior areas generating anticipatory signals that reach further into the future (Spearman's rho = 0.58, p=0.0030). Statistical thresholding was conducted via a permutation test, with correction for false discovery rate (FDR), q<0.05. (b) Event by time plots for three sample regions from (a), selected post hoc for illustration. Because the HMM produces a probability distribution across states at each timepoint, which can reflect a combination of current and upcoming event representations, we plot the expected value of the event assignments at each timepoint. The upward shift from the first viewing to subsequent viewings indexes the amount of anticipation.

*Figure 2 continued on next page*

*Figure 2 continued*

The online version of this article includes the following figure supplement(s) for figure 2:

**Figure supplement 1.** Unthresholded statistical map of anticipation timescales.
**Figure supplement 2.** Timescales of anticipation when the first viewing is compared to the last viewing.
**Figure supplement 3.** Timescales of anticipation as a function of the optimal number of events.
**Figure supplement 4.** Cross-correlation analysis of anticipation.

timecourse with the event annotation timecourse to find the temporal offset that maximized this correlation.

We found three clusters in the middle temporal gyrus (MTG), fusiform gyrus (FG), and superior temporal sulcus (STS) in which the optimal lag for the repeated viewings was significantly earlier than for the initial viewing, indicating that the relationship between the brain-derived HMM event boundaries and the human-annotated boundaries was changing with repeated viewings (*Figure 5*). The HMM boundaries on the first viewing were significantly later than the annotated boundaries in FG and STS, while the optimal lag did not significantly differ from 0 in MTG (95% confidence intervals for the optimal lag, in seconds: MTG = [−0.27, 2.86]; FG = [0.14, 1.99]; STS = [1.48, 8.53]). The HMM boundaries on repeated viewings were significantly earlier than the annotated boundaries in all three regions (95% confidence intervals for the average optimal lag, in seconds: MTG = [−4.06, −1.83]; FG = [−1.56, –0.26]; STS = [−3.06, –1.69]).

## Discussion

We investigated whether the brain contains a hierarchy of anticipatory signals during passive viewing of a naturalistic movie. We found that regions throughout the brain exhibit anticipation of upcoming events in audiovisual stimuli, with activity patterns shifting earlier in time as participants repeatedly watched the same movie clip. This anticipation occurred at varying timescales along the cortical hierarchy. Anticipation in higher-order, more anterior regions reached further into the future than that in lower-order, more posterior regions. Furthermore, in a subset of these regions, the coupling between event representations and human-annotated events shifted with learning: event boundaries in the brain came to reliably precede subjective event boundaries in the movie.

### Regions with anticipatory representations

One region showing long-timescale anticipatory signals was the bilateral anterior insula. This region has been linked to anticipation of diverse categories of positive and negative outcomes (*Liu et al., 2011*), including outcomes that will be experienced by other people (*Singer et al., 2009*). The movie stimulus used in our experiment depicts an interview in which the protagonist is initially judged to have 'zero' experience but then ends up impressing the interviewer, allowing for anticipation of this unexpected social outcome only on repeat viewings. Other regions showing long timescales of anticipation include the medial prefrontal cortex (mPFC), which tracks high-level narrative schemas (*Baldassano et al., 2018*) and has been proposed to play a general role in event prediction (*Alexander and Brown, 2014*), and lateral prefrontal cortex, including the inferior frontal gyrus, which processes structured sequences across multiple domains (*Uddén and Bahlmann, 2012*).

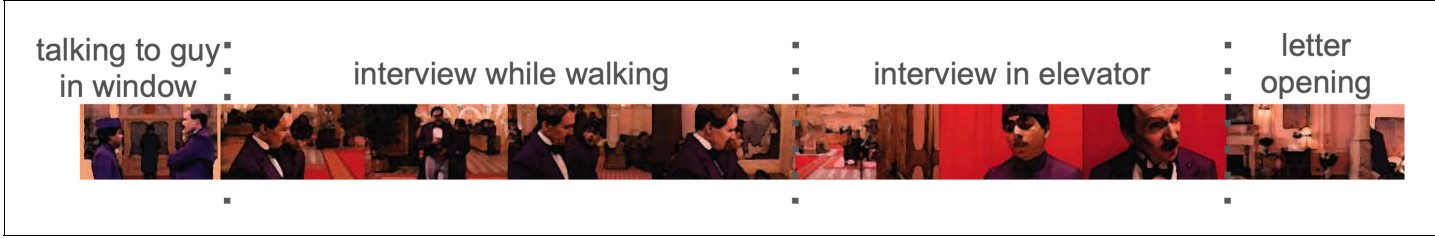

**Figure 3.** An example of event annotations from *The Grand Budapest Hotel*. Dotted lines demarcate events and phrases between the lines are brief titles given by one participant to describe each event. (Frames in this figure have been blurred to comply with copyright restrictions, but all participants were presented with the original unblurred version.)

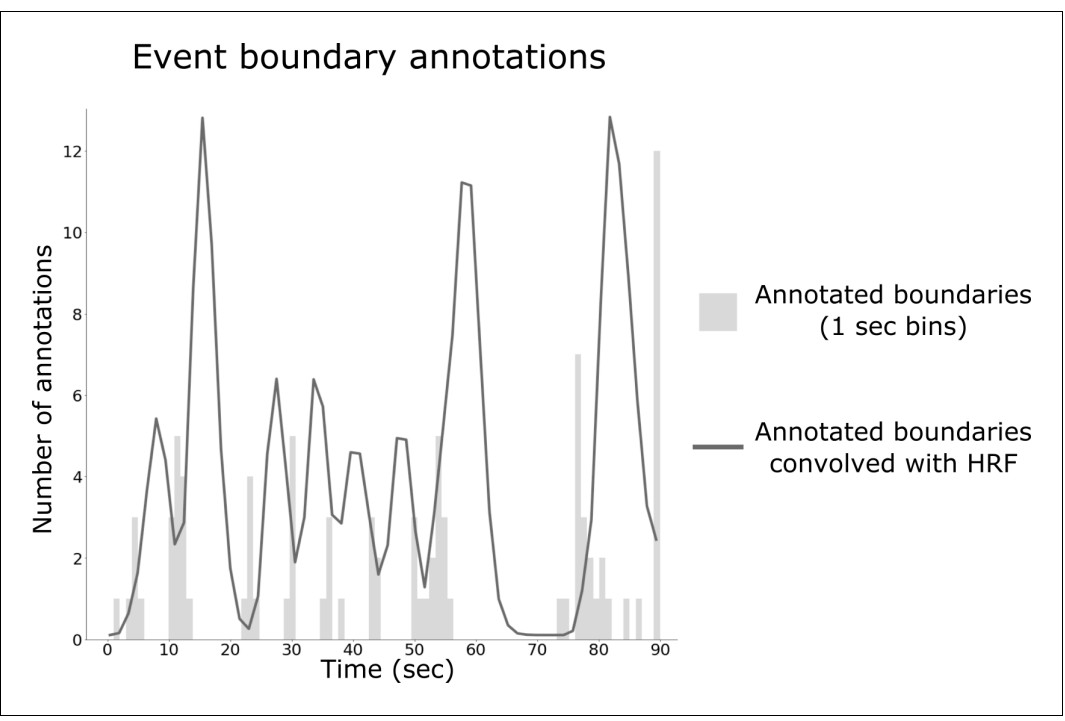

**Figure 4.** Construction of behavioral boundary timecourse from human annotations. The number of boundary annotations at each second of the movie clip (in gray) was convolved with a hemodynamic response function (HRF) to produce a continuous measure of boundary strength (black line).

We also observed shorter-timescale anticipation throughout lateral occipital and ventral temporal cortex, which, though primarily thought to process bottom-up visual information, also exhibits event-specific patterns during recall (*Chen et al., 2017*). A top-down memory-driven signal could be responsible for driving anticipatory activation in these regions during repeated movie viewing (*Finnie et al., 2021*). Future work incorporating eye-tracking measurements could determine whether anticipatory eye movements can account for the temporal shifts in these regions, or if this anticipation is separate from the representation of the current retinal input.

We did not observe widespread anticipatory signals in primary sensory areas, although some prior fMRI studies have been able to observe such signals in early regions such as V1 (*Alink et al., 2010*; *Ekman et al., 2017*; *Hindy et al., 2016*; *Kok et al., 2012*). One possibility is that the rich, ongoing sensory input dominated relatively small anticipatory signals in these regions. Paradigms involving periods without any sensory input (e.g., occasionally removing the audiovisual movie from the screen during repeated viewings) may be necessary to detect these subtle signals. Alternatively, ultra-fast fMRI sequences (*Ekman et al., 2017*) or alternative imaging modalities (discussed below) may be required to track anticipation at a subsecond scale.

## Relationship to previous studies of timescale hierarchies

Previous work has identified cumulatively longer timescales up the cortical hierarchy but has primarily focused on representations of the past. *Lerner et al., 2011* demonstrated hierarchical cortical dynamics in participants who listened to variants of a 7 min narrative that was scrambled at different timescales (e.g., paragraphs, sentences, or words). Response reliability, measured as the correlation in BOLD activity timecourses across individuals, varied based on the timescale of scrambling, with higher-level brain regions responding consistently to only the more-intact narrative conditions. This led to the idea that higher-order brain regions contain larger 'temporal receptive windows' than lower-order areas, in that their activity at a given moment is influenced by relatively more of the past. Likewise, using intracranial EEG (iEEG), *Honey et al., 2012* observed progressively longer temporal receptive windows in successive stages of the cortical hierarchy in participants who watched intact and scrambled versions of the movie *Dog Day Afternoon*. These findings can be described by

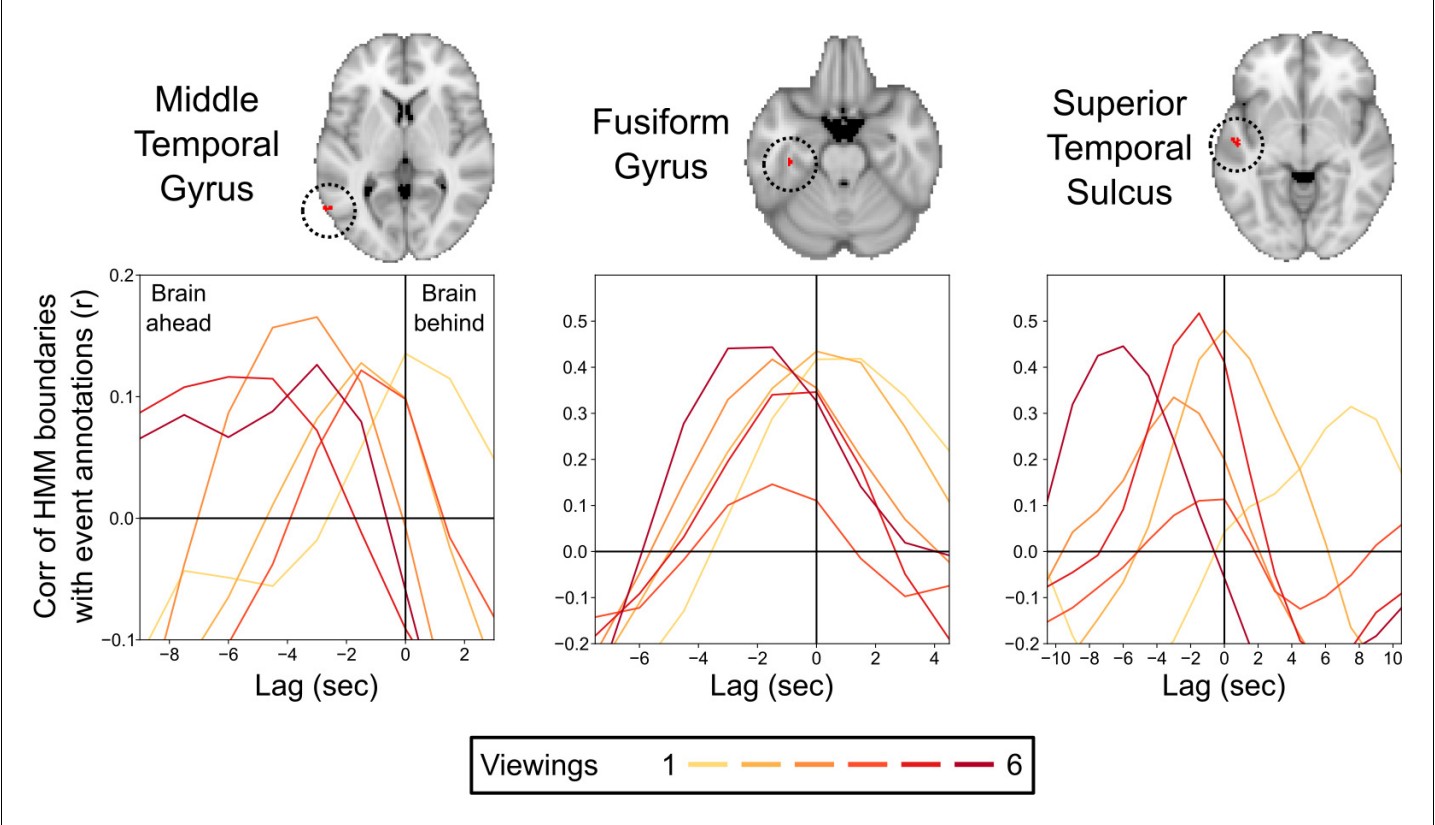

**Figure 5.** Correlations between the brain's event transitions and human-annotated event boundaries. Cross-correlation plots show the correlation between the brain's hidden Markov model (HMM) event boundaries and annotated event boundaries as the timecourses are shifted with respect to one another. The correlation at 0 lag indicates the similarity between the brain's event boundaries and annotated event boundaries when the timecourses are aligned. Negative lags show the correlations when the human-annotated event timecourse is shifted earlier in time, and positive lags show the correlation when the human-annotated event timecourse is shifted later in time. Peaks in the cross-correlation plot indicate the lag that produced the highest correlation between the brain's event boundaries and annotated event boundaries. On initial viewing, the HMM event boundaries for the fusiform gyrus and superior temporal sulcus lagged significantly behind the annotated event boundaries, while the timing of the peak correlation for the middle temporal gyrus did not significantly differ from 0 lag. On subsequent viewings, the HMM event boundaries in all three regions shifted to be significantly earlier than the initial viewing, with the timing of the peak correlation significantly preceding 0 lag.

the process memory framework (*Hasson et al., 2015*), where hierarchical memory timescales process, represent, and support longer and longer units of information. We found that this hierarchy also exists in the prospective direction, with the degree of anticipatory temporal shifts increasing from posterior-to-anterior regions of the brain. Furthermore, regions with longer intrinsic processing timescales showed further-reaching anticipation. These results extend the process memory framework, suggesting that the timescales in these regions are relevant not only for online processing and memory, but also for future anticipation or simulation.

Although prior work has uncovered anticipatory and predictive coding in the brain, most studies have examined fixed, shorter timescales of anticipation. Moreover, these shorter timescales have often been studied using simple, non-narrative stimuli such as objects moving across the screen, short visual sequences, and visual pattern completion tasks (*Alink et al., 2010*; *Ekman et al., 2017*; *Gavornik and Bear, 2014*; *Hindy et al., 2016*; *Kok et al., 2012*). Some studies have used dynamic movie stimuli, but anticipation was measured via correlations between initial and repeated viewing of a movie at a constant fixed lag of 2 s (*Richardson and Saxe, 2020*). Such an approach is not well suited to capturing dynamic levels of anticipation within and across brain regions.

Research investigating longer timescales of anticipation, such as learning future state representations in a maze task, examined single timescales up to 30 s ahead in OFC-VMPFC regions (*Elliott Wimmer and Büchel, 2019*). Some studies that use narrative stimuli have examined specially constructed texts in order to manipulate predictions about upcoming sentences; for

example, work by *Kandylaki et al., 2016* demonstrated that predictive processing of referents in narratives can be modulated by voice (passive vs. active) and causality (high vs. low). Our results show that in a naturalistic setting, in which structure exists at many timescales, anticipation at multiple levels can occur in parallel across different brain regions. We found anticipation up to approximately 15 s into the future with our 90 s stimulus, but future work with stimuli of longer duration could uncover even longer timescales of anticipation, on the scale of minutes. Simultaneously maintaining expectations at varying timescales could allow for flexible behaviors, because different timescales of anticipation may be helpful for a variety of tasks and actions. Taking action to avoid immediate harm or danger would require shorter timescales of prediction, whereas cultivating social relationships demands predictions on longer timescales.

These results are consistent with those of *Baldassano et al., 2017*, in which some participants listening to an audio narrative had advance knowledge of the high-level events of the story (because they had previously watched a movie version of the narrative). Using a similar HMM approach as in this paper, the authors observed shifts in event boundaries in higher-level regions including angular gyrus, posterior medial cortex, and mPFC. In the current study, however, participants were repeatedly exposed to an identical movie stimulus, allowing them to generate expectations at a broad range of timescales, including the timescales of fast-changing low-level visual features. This novel approach allowed us to observe for the first time that anticipation occurs in both low- and high-level regions, with shorter-timescale anticipation in visual occipital regions and the furthest-reaching anticipatory signals in prefrontal cortex.

Our model detects anticipation as temporal shifts in events, and though timepoints can reflect 'mixed' event assignments, it assumes that the underlying event patterns themselves (*Figure 1b*) are constant. This view of anticipation is complementary to other theories of predictive representations, in which event patterns themselves should change over time to incorporate future information. One example is the 'successor representation' model from the field of reinforcement learning, which describes a representation in which each state (here, event representation) comes to include features of future events, weighted by their likelihood of occurring and their distance into the future (*Dayan, 1993*). Successor representations can also be constructed at multiple scales (by changing the relative weighting of events near vs. far in the future). Such multi-scale representations are useful for goal-directed prediction that require multiple stages of planning (*Momennejad and Howard, 2018*; *Brunec and Momennejad, 2019*). Future work could explore how these two different theories could be integrated to model both mixing of event patterns and temporal shifts in the activation of these event patterns.

## Anticipation in other neuroimaging modalities

The current fMRI study is complementary to investigations of memory replay and anticipation that use MEG and iEEG. In an MEG study, *Michelmann et al., 2019* found fast, compressed replay of encoded events during recall, with the speed of replay varying across the event. Furthermore, an iEEG investigation found anticipatory signals in auditory cortex when individuals listened to the same story twice (*Michelmann et al., 2020*). In another MEG study, *Wimmer et al., 2020* found compressed replay of previously encoded information. Replay was forward when participants were remembering what came after an event, and backward when participants were remembering what came before an event. The forward replay observed in the Wimmer et al. study may be similar to the anticipatory signals observed in the current study, although there was no explicit demand on memory retrieval in our paradigm. Thus, one possibility is that the anticipatory signals observed in MEG or iEEG are the same as those we observe in fMRI, except that they are necessarily sluggish and smoothed in time when measured via a hemodynamic response. This possibility is supported by fMRI work showing evidence for compressed anticipatory signals, albeit at a slower timescale relative to MEG (*Ekman et al., 2017*).

An alternative possibility is that the anticipatory signals measured in our study are fundamentally different from those captured via MEG or iEEG. That could explain why we failed to find widespread anticipatory signals in primary visual or primary auditory cortex: the anticipatory signals in those regions might have been too fast to be captured with fMRI, particularly when competing with incoming, dynamic perceptual input. Future studies that obtain fMRI and MEG or iEEG in participants watching the same movie would be informative in that regard. It is possible that fMRI may be particularly well suited for capturing relatively slow anticipation of stable events, as opposed to faster

anticipatory signals relating to fast sub-events. Nevertheless, advances in fMRI analyses may allow the detection of very fast replay or anticipation, closing the gap between these methods and allowing more direct comparisons (*Wittkuhn and Schuck, 2021*).

## Future directions and conclusions

One limitation of the current work is the reliance on one movie clip. Movie clips of different durations might yield different results. For example, it is an open question whether the duration of anticipation scales with the length of the movie and playback speed or if the amount of anticipation is fixed (*Lerner et al., 2014*; *Baumgarten et al., 2021*). Furthermore, the content of the movie and how frequently event boundaries occur may change anticipation amounts. That said, anticipatory signals in naturalistic stimuli have been observed across multiple studies that use different movies and auditorily presented stories (e.g., *Baldassano et al., 2017*; *Michelmann et al., 2020*; also see *Michelmann et al., 2019*; *Elliott Wimmer and Büchel, 2019*; *Wimmer et al., 2020*). Thus, it is likely that anticipatory hierarchies will also replicate across different stimuli. There may nevertheless be important differences across stimuli. For example, the specific regions that are involved in anticipation may vary depending on what the most salient features of a movie or narrative are (e.g., particular emotional states, actions, conversations, or perceptual information).

The detection of varying timescales of anticipation in the brain can be applied to multiple domains and modalities of memory research. Future work could explore even shorter timescales using other neuroimaging modalities, or longer timescales using longer movies or narratives from TV series that span multiple episodes. Furthermore, the impact of top-down goals on the hierarchy of anticipation timescales could be explored by using different tasks that require different levels of anticipation, such as anticipating camera angle changes vs. location changes. Brain stimulation studies or studies of patients with brain lesions could also explore the extent to which anticipation in lower-level regions relies on feedback from higher-level regions (*Auksztulewicz and Friston, 2016*; *Kiebel et al., 2008*).

The increased use of naturalistic, dynamic stimuli in neuroscience, and the development of methods to analyze the resulting data, has opened many avenues for research exploring flexible, future-oriented behavior. Our results and analysis approach provide a new framework for studying how anticipatory signals are distributed throughout the cortex, modulated by prior memory, and adaptive for improving comprehension and behavior.

## Materials and methods

### Grand Budapest Hotel dataset

We used data collected by *Aly et al., 2018*. Thirty individuals (12 men, age: M = 23.0 years, SD = 4.2; education: M = 15.3 years, SD = 3.2; all right-handed) watched movie clips from *The Grand Budapest Hotel* while undergoing fMRI. None of the participants reported previously seeing this movie. We analyzed data from the *Intact* condition, during which participants watched a continuous 90 s clip from the movie in its original temporal order. This clip was watched six times, interspersed with other video clips that are not considered here. This *Intact* clip depicts an interview scene between the protagonist and his future employer inside of the Grand Budapest Hotel. Stimuli and data are available on OpenNeuro: https://openneuro.org/datasets/ds001545/versions/1.1.1.

Data were acquired on a 3T Siemens Prisma scanner with a 64-channel head/neck coil using a multiband echo planar imaging (EPI) sequence (repetition time = 1.5 s; echo time = 39 ms; flip angle = 50°; acceleration factor = 4; shift = 3; voxel size = 2.0 mm iso). T1-weighted structural images (whole-brain high-resolution; 1.0 mm iso) were acquired with an MPRAGE sequence. Field maps (40 oblique axial slices; 3 mm iso) were collected to aid registration. The fMRI scan took place over three experimental runs, each of which contained two presentations of the *Intact* movie clip (as well as other movie clips not considered here).

The first three EPI volumes of each run were discarded to allow for T1 equilibration. Data preprocessing was carried out in FSL, and included brain extraction, motion correction, high-pass filtering (max period = 140 s), spatial smoothing (3 mm FWHM Gaussian kernel), and registration to standard Montreal Neurological Institute (MNI) space. After preprocessing, the functional images for each run were divided into volumes that corresponded to each of the video clips presented within that run,

and only the two *Intact* clips within each run are considered further. Finally, each voxel's timecourse was z-scored to have zero mean and unit variance.

## Event annotations by human observers

Fourteen individuals (nine men) were asked to mark event boundaries corresponding to the same 90 s *Intact* clip from *The Grand Budapest Hotel* as shown to the fMRI participants. Each participant was asked to pause the clip at the end of a meaningful segment and to record the time and a brief title corresponding to the segment (*Figure 3*). Specifically, they were given the following instructions: *The movie clip can be divided into meaningful segments. Record the times denoting when you feel like a meaningful segment has ended. Pause the clip at the end of the segment, write down the time in the spreadsheet, and provide a short, descriptive title. Try to record segments with as few viewings of the movie clip as possible; afterward, record the number of times you viewed the clip.* Although participants were allowed to watch the clip multiple times, they were instructed to minimize and report the number of viewings needed to complete the task. No participant reported watching the clip more than three times.

## Detecting anticipatory signals using an event segmentation model

Group-averaged fMRI data were fit with the event segmentation model described by *Baldassano et al., 2017*. This HMM assumes that (1) events are a sequence of discrete states, (2) each event is represented in the brain by a unique spatial activity pattern, and (3) all viewings of the movie evoke the same sequence of activity patterns in the same order (though possibly with different timings). We fit the HMM jointly to all six viewings. This fitting procedure involved simultaneously estimating a sequence of event activity patterns that were shared across viewings, and estimating the probability of belonging to each of these events for every timepoint in all six datasets. The model was fit with seven events; this number was chosen to match the approximate timescale of the semantic events in the narrative, matching the mean number of events annotated by human observers (mean = 6.5).

After fitting the HMM, we obtain an event by timepoint matrix for each viewing, giving the probability that each timepoint belongs to each event. Note that because this assignment of timepoints to events is probabilistic, it is possible for the HMM to detect that the pattern of voxel activity at a timepoint reflects a mixture of multiple event patterns. This allows us to track subtle changes in the timecourse of how the brain is transitioning between events. We took the expectation over events at each timepoint, yielding curves showing the average event label at each timepoint for each viewing. To compute shifts in time between the first viewing and the average of repeated viewings, the area under the curve (AUC) was computed for each viewing. We then computed the amount of anticipation as the average AUC for repeated viewing (viewings 2–6) minus the AUC for the first viewing. In a supplementary analysis, we compared the first viewing to the last viewing only. To convert to seconds, we divide by the vertical extent of the graph (number of events minus 1) and multiplied by the repetition time (1.5 s). We then performed a one-tailed statistical test (described below) to determine whether this difference was significantly positive, indicating earlier event transitions with repeated viewing. Not only does this approach provide a way of quantifying anticipation, it gives us a trajectory of the most likely event at any given timepoint, as well as the onset and duration of each event.

We obtained whole-brain results using a searchlight analysis. We generated spherical searchlights spaced evenly throughout the MNI volume (radius = 5 voxels; stride = 5 voxels). We retained only the searchlights with at least 20 voxels which were inside a standard MNI brain mask and for which at least 15 participants had valid data for all viewings. We then used the SRM (*Chen et al., 2015*) to functionally hyperalign all participants into shared 10-dimensional space (jointly fitting the alignment across all six viewings) and averaged their responses together. This produced a 10 feature by 60 timepoint data matrix for each of the six viewings, which was input to the HMM analysis described above. After running the analysis in all searchlights, the anticipation in each voxel was computed as the average anticipation of all searchlights that included that voxel.

To assess statistical significance, we utilized a permutation-based null hypothesis testing approach. We constructed null datasets by randomly shuffling each participant's six responses to the six presentations of the movie clip. The full analysis pipeline (including hyperalignment) was run 100

times, once on the real (unpermuted) dataset and 99 times on null (permuted) datasets, with each analysis producing a map of anticipation across all voxels. A one-tailed p-value was obtained in each voxel by fitting a normal distribution to the null anticipation values, and then finding the fraction of this distribution that exceeded the real result in this voxel (i.e., showed more anticipation than in our unpermuted dataset). Voxels were determined significant ($q<0.05$) after applying the Benjamini-Hochberg FDR correction, as implemented in AFNI (**Cox, 1996**).

To determine if anticipation systematically varied across the cortex in the hypothesized posterior-to-anterior direction, we calculated the Spearman's correlation between the Y-coordinate of each significant ($q<0.05$) voxel (indexing the position of that voxel along the anterior/posterior axis) and the mean amount of anticipation in that voxel. To obtain a p-value, the observed correlation was compared to a null distribution in which the Spearman's correlation was computed with the null anticipation values from the permutation analysis described above, in which the order of the view-ings was randomly scrambled for each participant. For comparison, the correlation was also com-puted for the X (left-right) and Z (inferior-superior) axes. This analysis was repeated on unthresholded anticipation maps, to examine if this hierarchy remained even when including regions whose anticipation amounts did not reach statistical significance.

To relate the timescales of anticipation to the intrinsic timescales of brain regions during the first viewing, we fit the HMM on the first viewing alone, varying the number of events from 2 to 10. The HMM was trained on the average response from half of the participants (fitting the sequence of activity patterns for the events and the event variance) and the log-likelihood of the model was then measured on the average response in the other half of the participants. The training and testing sets were then swapped, and the log-likelihoods from both directions were averaged together. Hypera-lignment was not used during this fitting process, to ensure that the training and testing sets remained independent. The number of events that yielded the largest log-likelihood was identified as the optimal number of events for that searchlight. The optimal number of events was then com-pared to the anticipation timescale in that region (from the main analysis), using Spearman's correlation.

For comparison, we also ran a searchlight looking for anticipatory effects using a non-HMM cross-correlation approach. Within each searchlight, we obtained an average timecourse across all voxels and correlated the response to the first viewing with the average response to repeated viewings at differing lags. Using the same quadratic-fit approach for identifying the optimal lag described below, we tested whether the repeated-viewing timecourse was significantly ahead of the initial-viewing timecourse (relative to a null distribution in which the viewing order was shuffled within each sub-ject). The p-values obtained were then corrected for FDR.

## Comparison of event boundaries in brain regions to annotations

We compared the event boundaries identified by the HMM within each searchlight to the event boundaries annotated by human observers. To obtain an event boundary timecourse from the anno-tations, we convolved the number of annotations (across all raters) at each second with the HRF (**Fig-ure 4**). Separately, we generated a continuous measure of HMM 'boundary-ness' at each timepoint by taking the derivative of the expected value of the event assignment for each timepoint, as illus-trated in **Figure 1d**. Moments with high boundary strength indicate moments in which the brain pat-tern was rapidly switching between event patterns. We cross-correlated the HMM boundary strength timecourse for each viewing with the annotated event boundary timecourse, shifting the annotated timecourse forward and backward to determine the optimal temporal offset (with the highest correlation). We measured the timing of the peak correlation by identifying the local maxi-mum in correlation closest to 0 lag, then fitting a quadratic function to the maximum correlation lag and its two neighboring lags and recording the location of the peak of this quadratic fit. This pro-duced a continuous estimate of the optimal lag for each viewing. We measured the amount of shift between the optimal lag for the first viewing and the average of the optimal lags for repeated view-ings, and obtained a p-value by comparing to the null distribution over maps with permuted viewing orders (as in the main analysis), then performed an FDR correction.

We identified three gray matter clusters significant at $q<0.05$. To statistically assess whether the optimal lags differed from 0 in the three searchlights maximally overlapping these three clusters, we repeated the cross-correlation analysis in 100 bootstrap samples, in which we resampled from the raters who generated the annotated event boundaries. We obtained 95% bootstrap confidence

intervals for maximally correlated lag on the first viewing and for the average of the maximally correlated lags on repeated viewings.

## Code and resource availability

Data preprocessing scripts and python code to reproduce all the results in this paper are available at https://github.com/dpmlab/Anticipation-of-temporally-structured-events (copy archived at swh:1:rev:8fbd488c04d47148f9a53048de5d05a90e1c1663). Results in MNI space can be viewed at https://identifiers.org/neurovault.collection:9584.

## Acknowledgements

We thank the Aly and Baldassano labs for their feedback and support during this project, Janice Chen for helpful conversations about prediction hierarchies, and our three reviewers for proposing many useful improvements to the analyses.

## Additional information

### Funding

No external funding was received for this work.

### Author contributions

Caroline S Lee, Software, Formal analysis, Validation, Investigation, Visualization, Writing - original draft, Project administration, Writing - review and editing; Mariam Aly, Conceptualization, Supervision, Writing - original draft, Writing - review and editing; Christopher Baldassano, Conceptualization, Software, Formal analysis, Supervision, Validation, Visualization, Methodology, Writing - original draft, Writing - review and editing

### Author ORCIDs

Caroline S Lee 🔳 https://orcid.org/0000-0002-7769-8799
Mariam Aly 🔳 http://orcid.org/0000-0003-4033-6134
Christopher Baldassano 🔳 https://orcid.org/0000-0003-3540-5019

### Decision letter and Author response

Decision letter https://doi.org/10.7554/eLife.64972.sa1
Author response https://doi.org/10.7554/eLife.64972.sa2

## Additional files

### Supplementary files

• Transparent reporting form

### Data availability

We used a publicly-available dataset, from https://openneuro.org/datasets/ds001545/versions/1.1.1.

The following previously published dataset was used:

| Author(s) | Year | Dataset title | Dataset URL | Database and Identifier |
|---|---|---|---|---|
| Aly M, Chen J, Turk-Browne NB, Hasson U | 2019 | Learning Naturalistic Temporal Structure in the Posterior Medial Network | https://openneuro.org/datasets/ds001545/versions/1.1.1 | OpenNeuro, 10.18112/openneuro.ds001545.v1.1.1 |

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
