## [Decision Letter]

**Acceptance summary:**

This study uses innovative fMRI analysis methods to show how the brain predicts the future. It provides compelling evidence for anticipatory neural activity during repeated viewing of a movie clip, finding that different brain regions anticipate events to different degrees, mirroring the temporal integration windows of these brain regions.

**Decision letter after peer review:**

Thank you for submitting your article "Anticipation of temporally structured events in the brain" for consideration by eLife. Your article has been reviewed by 3 peer reviewers, one of whom is a member of our Board of Reviewing Editors, and the evaluation has been overseen by Timothy Behrens as the Senior Editor. The following individual involved in review of your submission has agreed to reveal their identity: Linda Geerligs (Reviewer #2).

Essential Revisions:

1. Provide more statistical support for differences between brain regions and the anterior-posterior hierarchy

2. Analyze repeated viewings separately, also to account for differences in reliability

3. Reconsider the event correlation analysis (see several specific suggestions below)

4. Consider optimizing analyses to reduce noise (and false negatives)

*Reviewer #1 (Recommendations for the authors (required)):*

– Were the correlations in Figure 6 corrected for multiple comparisons?

– The introduction mentions that "The majority of studies examining anticipatory signals, however, have tested only one-step associations". Arguably, the current results might also reveal one-step associations, but with the step duration being longer in some regions than others (in line with previous findings of different temporal integration windows). Please discuss this possibility or clarify what you mean with one-step associations.

– The results are based on one 90-s movie segment with 7 events, mostly involving humans. To what extent may results be specific to this short segment and these specific events (e.g., anticipation of theory of mind or emotions)?

– To get a sense of the false-positive rate, it would be informative to see the same map (Figure 2) but testing for the opposite temporal direction, as Supplementary file.

– As Supp Figure 2, please also show unthresholded maps (cf Figure 3 of Baldassano, 2017) plotting both positive and negative anticipation. This would give a fuller insight into the data, also in regions that didn't cross the threshold.

– In the table in Figure 6, one correlation of 0.52 is not indicated as significant (cluster 7). Similarly, the difference (0.29) is also not indicated as significant. Is this correct?

*Reviewer #2 (Recommendations for the authors (required)):*

Below I will mention concrete suggestions for improvement related to the points in the public recommendation.

1. I would suggest repeating all analyses using the estimated real/optimal number of events in each brain region, rather than the number that was based on behavioural annotations.

2. Repeating the analyses with hyper-aligned data should reduce the amount of noise in the group-averaged data.

3. Using only viewing 6 as the repeated viewing condition may improve the detection of anticipatory signals in early sensory areas. Looking at how the amount of anticipation changes across all viewings would add an interesting new dimension to the results presented in the paper.

4. An alternative approach to this analysis is to vary the HRF delay for the annotated events and investigate which delay shows the optimal correlation. This approach would provide additional evidence for the estimated amount of anticipation shown in figure 2.

5. Rather than grouping voxels based on the identified cluster, I would suggest either sticking to the original searchlight definitions or grouping searchlights based on the similarity of their event boundaries.

6. A discussion about this issue may be a valuable addition to the Discussion section.

*Reviewer #3 (Recommendations for the authors (required)):*

1. The discussion is relatively quite long.

2. It seems like the brain maps in Figure 6 should be added to Figure 2, or their own figure, before the annotation correlation-related results in Figure 5 and the table in Figure 6. As presented, it is confusing and not initially clear why there are clusters with no significant correlation results – that the annotation analysis presented is independent from the analysis that identified the clusters.

3. In Figure 5, the temporal language is unclear. 'Backward' and 'forward' here are confusing descriptors, e.g. backward can be behind the current position (earlier in time) or pushed 'back' later in time. On a different note about this figure, it should have brain labels in the text of the correlation plots, the same cluster numbering added as in Figure 6, and panel letters.

4. The original report in Aly et al., (2018) notes that no participants had previously viewed the movie that the clips were taken from. I may have missed it, but it would be helpful to repeat that information here.

---

## [Author Response]

Essential Revisions:1. Provide more statistical support for differences between brain regions and the anterior-posterior hierarchy

Thank you for raising this important concern. We briefly outline our major changes here, and describe our changes in more detail below in response to individual reviewer comments. First, we related anticipation amounts to the position of brain regions along the anterior/posterior axis, and indeed found that anticipation amounts progressively increase from posterior to anterior parts of the brain (Spearman’s rho = 0.58, p = 0.0030). Second, this systematicity exists even when the analysis was done on an unthresholded statistical map (Spearman’s rho = 0.42, p = 0.0028; Figure 2—figure supplement 1). Finally, we explored whether brain regions with faster vs slower activity dynamics (i.e., more vs fewer events during the initial viewing of the movie) showed differences in anticipation amounts. We found that regions that integrate information over more of the past (i.e., show fewer, longer events) show more anticipation (Spearman’s rho = 0.319, p = 0.00031; Figure 2 – Figure supplement 3). This is consistent with our hypothesis that the *retrospective* temporal hierarchy observed in prior studies (Hasson et al., 2008; Hasson et al., 2015; Lerner et al., 2011) is directly related to the anticipatory hierarchy that we describe here.

2. Analyze repeated viewings separately, also to account for differences in reliability

We implemented the changes recommended by the viewers. First, we entered each viewing separately into the HMM analyses and then averaged the amount of anticipation across repetitions 2-6 (rather than averaging the timecourses of the repetitions before entering them into the HMM). This is now our main analysis, and each viewing is now shown as a separate line in our example time by event plots (Figure 2B). We also compared the first viewing to the last (6th) viewing alone, which yielded a similar result (Figure 2—figure supplement 2). Finally, when relating the brain’s event representations to human-annotated events (Figure 5), we examine the brain data for each viewing separately and present the data for all six movie presentations. Our main conclusions remain unchanged when taking these approaches.

3. Reconsider the event correlation analysis (see several specific suggestions below)

We agree that the most important test for this analysis is whether there is a systematic shift, across movie repetitions, in the timing of the peak cross-correlation between the brain’s event transitions and human-annotated event boundaries. To test this, we conducted a new analysis in which we measured the timing of the peak cross-correlation between HMM-derived event transitions in the brain and the human-annotated event boundaries, separately for each of the six movie viewings. In other words, we found the amount of shift in the brain’s event transitions that led to the maximum correlation with the timing of the human-annotated event boundaries. We then compared the timing of the correlation peak for the first movie viewing to the timing of the mean peak across viewings 2-6, and found regions of the brain where the peak shifted to be earlier with subsequent movie viewings. This was done as a whole-brain analysis with FDR correction. We include a figure (Figure 5) showing the data for the three searchlights that corresponded to clusters that met the q <.05 FDR criterion.

The preceding analysis looked for regions for which the timing of the peak cross-correlation between the brain’s events and human-annotated events shifted earlier over movie repetitions, but did not test for the absolute location of that peak correlation (relative to zero lag between the HMM events and annotated events). Do the brain’s event transitions occur before annotated event transitions, after, or are they aligned? And how does this change over movie repetitions? We examined this question in the three clusters that emerged from the analysis in the preceding paragraph. We found that for the initial viewing, the brain’s event transitions lagged behind human-annotated event boundaries for two of the three clusters, whereas for the last cluster, the brain’s transitions and subjective event boundaries were aligned. For repeated viewings, the timing of the peak correlations shifted such that the brain’s representations of an event transition reliably preceded the occurrence of the human-annotated event boundary, for all three clusters (Figure 5).

Together, these results confirm that, in some regions, the best alignment between the brain’s event transitions and human-annotated event boundaries shifts over movie repetitions such that the brain’s event transitions start to occur earlier over repetitions. In particular, the brain’s events shift to precede subjective event boundaries.

4. Consider optimizing analyses to reduce noise (and false negatives)

We implemented many changes to this end, which we will briefly describe here and describe in more detail in response to individual reviewer comments. First, we hyperaligned the brain data of individual participants before conducting our anticipation analyses, using the "Shared Response Model" (SRM) hyperalignment approach (Chen et al., 2015). Hyperalignment projects features (e.g., voxels) from individual brains into a common high-dimensional space, in which features across individuals share functional properties as opposed to anatomical locations. This approach increases the sensitivity of analyses, such as our HMM approach, that depend on across-brain similarities because traditional anatomical alignment approaches do not accommodate idiosyncrasies in fine-grained functional topographies across individuals. This approach uncovered anticipation in more widespread regions compared to our initial (anatomically aligned) analyses. The hyperaligned analyses now replace our prior analyses. Second, due to reviewer concerns about whether we were properly controlling the rate of false positives in our maps, we replaced our bootstrapping-based approach with a permutation-based approach. Rather than resampling participants to produce confidence intervals on our results, we permuted the order of the viewings to generate null maps and then computed p values by comparing our results to these null results. This approach yielded similar p values to our original bootstrapping approach, verifying that we are appropriately controlling our false positive rate. We then applied a False Discovery Rate (FDR) correction as before, to account for multiple comparisons across voxels. Third, we now share and analyze unthresholded maps of anticipation in the brain. As noted above, the anticipation hierarchy persists even when the analysis is conducted on an anticipation map that was not corrected for statistical significance.

Reviewer #1 (Recommendations for the authors (required)):– Were the correlations in Figure 6 corrected for multiple comparisons?

In our initial manuscript, those correlations were not corrected for multiple comparisons. In our revision, we have replaced our initial analysis examining the relationship between the brain’s event transitions and human-annotated event boundaries with a new analysis. This new analyses tests for shifts in the relationship between brain and human-annotated event boundaries over movie viewings in a searchlight across the whole cortex (as described in Essential Revision # 3; see Figure 5). This new analysis is FDR-corrected at q < 0.05.

– The introduction mentions that "The majority of studies examining anticipatory signals, however, have tested only one-step associations". Arguably, the current results might also reveal one-step associations, but with the step duration being longer in some regions than others (in line with previous findings of different temporal integration windows). Please discuss this possibility or clarify what you mean with one-step associations.

Thank you for raising this issue. We have clarified our language, to state that most studies have used discrete items as stimuli and looked for anticipation of the single item that was coming up next:

“The majority of studies examining anticipatory signals, however, have tested anticipation based on memory for relatively simple associations between pairs of discrete stimuli, such as auditory tones , lines, dots, oriented gratings, or abstract objects (e.g., Alink, Schwiedrzik, Kohler, Singer, and Muckli, 2010; Gavornik and Bear, 2014; Hindy, Ng, and Turk-Browne, 2016; Kok, Jehee, and de Lange, 2012; Kok, Failing, and de Lange, 2014; Kok and Turk-Browne, 2018). These studies have found anticipatory signals about a single upcoming stimulus in a variety of brain regions, from perceptual regions (Kok et al., 2012, 2014) to the medial temporal lobe (Hindy et al., 2016; Kok and Turk-Browne, 2018).” (p.2)

– The results are based on one 90-s movie segment with 7 events, mostly involving humans. To what extent may results be specific to this short segment and these specific events (e.g., anticipation of theory of mind or emotions)?

Thank you for raising this point, which we agree is important. We now mention this limitation in the Discussion:

“One limitation of the current work is the reliance on one movie clip. Movie clips of different durations might yield different results. For example, it is an open question whether the duration of anticipation scales with the length of the movie or if the amount of anticipation is fixed (c.f., Lerner, Honey, Katkov, and Hasson, 2014). Furthermore, the content of the movie and how frequently event boundaries occur may change anticipation amounts. That said, anticipatory signals in naturalistic stimuli have been observed across multiple studies that use different movies and auditorily presented stories (e.g., Baldassano et al., 2017; Michelmann et al., 2020; also see Michelmann et al., 2019; Wimmer and Büchel, 2019; Wimmer et al., 2020). Thus, it is likely that anticipatory hierarchies will also replicate across different stimuli. There may nevertheless be important differences across stimuli. For example, the specific regions that are involved in anticipation may vary depending on what the most salient features of a movie or narrative are (e.g., particular emotional states, actions, conversations, or perceptual information).” (p.10)

– To get a sense of the false-positive rate, it would be informative to see the same map (Figure 2) but testing for the opposite temporal direction, as Supplementary file.

We agree that it is important to get a sense of the false-positive rate. We considered this proposed approach, but we are not sure if the opposite temporal direction provides a measure of the false positive rate. Instead, the opposite temporal direction — on repeated viewings, brain areas lag behind initial viewing — could be an interesting phenomenon in its own right. That could reflect, for example, holding on to the past for longer amounts of time in order to better integrate information with what is coming up next.

We therefore opted to use permutation tests to get a more direct measure of the false positive rate. For these permutation tests, we randomly shuffled movie viewings within each participant before conducting the anticipation analysis. This random shuffling was done 99 items, allowing us to obtain a null distribution of anticipation for each searchlight. This null distribution was used to calculate a p-value for each searchlight by computing the z-score of our result relative to the null distribution and then obtaining a p-value from a Normal survival function. The p-value map was then FDR-corrected with q < 0.05.

Although we did not statistically test for temporal shifts in the opposite direction (with activity shifting later on repeated viewings), the unthresholded map of positive and negative anticipation can be viewed at https://identifiers.org/neurovault.collection:9584, and shows very few regions with negative values of anticipation.

– As Supp Figure 2, please also show unthresholded maps (cf Figure 3 of Baldassano, 2017) plotting both positive and negative anticipation. This would give a fuller insight into the data, also in regions that didn't cross the threshold.

We agree this is useful, and have included the unthresholded map as Figure 2—figure supplement 1. As noted earlier, the posterior-to-anterior hierarchy of anticipation is also present in this unthresholded map.

– In the table in Figure 6, one correlation of 0.52 is not indicated as significant (cluster 7). Similarly, the difference (0.29) is also not indicated as significant. Is this correct?

That was actually correct; that value failed to reach statistical significance because of very high variance. That said, that analysis (and table) has since been replaced (see Essential Revision #3).

Reviewer #2 (Recommendations for the authors (required)):Below I will mention concrete suggestions for improvement related to the points in the public recommendation.1. I would suggest repeating all analyses using the estimated real/optimal number of events in each brain region, rather than the number that was based on behavioural annotations.

Please see our response under point #2 in the public recommendation of this reviewer. Briefly, we found that the optimal number of events (defined on the first viewing) did differ across the cortex in a way that was correlated with the degree of anticipation. We chose to use a fixed number of events in the anticipation analyses in the manuscript to avoid a potential confound between the number of events used in the HMM and the degree of anticipation detected.

That said, we nevertheless ran this proposed analysis, in which the number of events used in the anticipation analysis was set to the optimal value based on the first viewing in each region, and obtained the following result (thresholded at q<0.05):

Although noisier than our main analysis, this result does replicate the general posterior-to-anterior topography of anticipation. Again, we chose not to include this in the manuscript due to our concerns that this analysis could produce an artificial relationship between event timescales during initial viewing and degree of anticipation. We hope that our compromise — relating optimal event numbers to anticipation amounts in our main analysis — is a satisfactory approach given the difficulties in interpreting this proposed analysis.

2. Repeating the analyses with hyper-aligned data should reduce the amount of noise in the group-averaged data.

This useful suggestion has been implemented. All the analyses were repeated after hyper-alignment. The same pattern of results emerged, but anticipatory signals are now generally more widespread and robust.

3. Using only viewing 6 as the repeated viewing condition may improve the detection of anticipatory signals in early sensory areas. Looking at how the amount of anticipation changes across all viewings would add an interesting new dimension to the results presented in the paper.

Thank you for these recommendations. We now statistically compare the first viewing to the last viewing and show those results as Figure 2—figure supplement 2. These data exhibit the same overall pattern as the first viewing compared to all subsequent viewings. We also plot data for each viewing separately in the event by time plots in Figure 2. Finally, we show the relationship between the brain’s event boundaries and human-annotated event boundaries for each viewing separately in Figure 5. Visual inspection of the latter two figures shows that anticipation generally increases with subsequent movie viewings.

4. An alternative approach to this analysis is to vary the HRF delay for the annotated events and investigate which delay shows the optimal correlation. This approach would provide additional evidence for the estimated amount of anticipation shown in figure 2.

Thank you for this suggestion. If we are interpreting it correctly, this is functionally equivalent to what we did. In particular, a reduced HRF delay ( i.e., an HRF that is shifted earlier in time) is analogous to shifting the convolved timecourse earlier in time. Likewise, an increased HRF delay (i.e., an HRF that is shifted later in time) is analogous to shifting the convolved timecourse later in time. However, please let us know if we misinterpreted your comment and should consider a different alternative. For example, if the suggestion is to change the delay between the HRF onset and its peak, that would indeed yield different results. However, such an analysis would have to be done carefully so that it remains biologically plausible.

5. Rather than grouping voxels based on the identified cluster, I would suggest either sticking to the original searchlight definitions or grouping searchlights based on the similarity of their event boundaries.

All analyses are now conducted with the same whole-brain searchlight approach, without any post-hoc grouping or clustering.

6. A discussion about this issue may be a valuable addition to the Discussion section.

We have clarified that the HMM produces a probability distribution across states (events) at each time-point, i.e., an activity pattern at any given time-point can reflect a mixture of current and upcoming events.

This is clarified in the caption to Figure 1:

“By fitting a Hidden Markov Model (HMM) jointly to all viewings, we can identify this shared sequence of event patterns, as well as a probabilistic estimate of event transitions. Regions with anticipatory representations are those in which event transitions occur earlier in time for repeated viewings of a stimulus compared to the initial viewing , indicated by an upward shift on the plot of the expected value of the event at each timepoint.” (p.3)

And in the caption to Figure 2:

“Because the HMM produces a probability distribution across states at each timepoint, which can reflect a combination of current and upcoming event representations, we plot the expected value of the event assignments at each timepoint.” (p.5)

And also in the Methods:

“After fitting the HMM, we obtain an event by time-point matrix for each viewing , giving the probability that each timepoint belongs to each event. Note that, because this assignment of timepoints to events is probabilistic, it is possible for the HMM to detect that the pattern of voxel activity at a timepoint reflects a mixture of multiple event patterns, allowing us to track subtle changes in the timecourse of how the brain is transitioning between events.” (p.13)

Reviewer #3 (Recommendations for the authors (required)):1. The discussion is relatively quite long.

We apologize for that! We tried to condense when possible, but it was unfortunately difficult given reviewer requests to relate our findings to other relevant work and clarify the implications of our results. We have now added subsection headings to better organize the Discussion.

2. It seems like the brain maps in Figure 6 should be added to Figure 2, or their own figure, before the annotation correlation-related results in Figure 5 and the table in Figure 6. As presented, it is confusing and not initially clear why there are clusters with no significant correlation results – that the annotation analysis presented is independent from the analysis that identified the clusters.

We agree that running the annotation correlation analysis on post-hoc clusters led to some confusion. In the revised version of the manuscript, the (new) annotation correlation analysis is conducted as a separate whole-brain searchlight analysis, and the significant clusters are shown in Figure 5.

3. In Figure 5, the temporal language is unclear. 'Backward' and 'forward' here are confusing descriptors, e.g. backward can be behind the current position (earlier in time) or pushed 'back' later in time. On a different note about this figure, it should have brain labels in the text of the correlation plots, the same cluster numbering added as in Figure 6, and panel letters.

Thank you for pointing this out. We changed that sentence to the following:

“Negative lags show the correlations when the human-annotated event timecourse is shifted earlier in time, and positive lags show the correlation when the human-annotated event timecourse is shifted later in time.” (p.7)

With respect to Figure 5, the old figure has been replaced with one depicting the analyses in which we look for significant *shifts* in the peak cross-correlation between the brain’s event boundaries and human-annotated event boundaries. The new figure has the clusters labeled clearly with their names, and we no longer use cluster numbers (see discussion with reviewer #1, with respect to functionally heterogeneous clusters).

4. The original report in Aly et al., (2018) notes that no participants had previously viewed the movie that the clips were taken from. I may have missed it, but it would be helpful to repeat that information here.

This is a good point, and we have added this clarification to the Methods:

“None of the participants reported previously seeing this movie.” (p.12)